# Comparison of the Nonlinear Dynamic Pre- and Post-LED Equalization

**DOI:** 10.3390/s22051782

**Published:** 2022-02-24

**Authors:** Jerzy Siuzdak

**Affiliations:** Institute of Telecommunications, Warsaw University of Technology, Nowowiejska 15/19, 00-665 Warsaw, Poland; siuzdak@tele.pw.edu.pl

**Keywords:** visible light communications, LED, dynamic nonlinearity, equalization

## Abstract

Visible Light Communications (VLC) have gained much popularity lately. In such a system, a white LED (Light-Emitting Diode) plays a double role as a light source and a transmitter. The main problem here is that the LED exhibits a low bandwidth and high nonlinearity, so the equalization of the LED nonlinear dynamic response is necessary. For this, various equalizers are used. This paper compares the pre- and post-equalizer performance in terms of the received signal quality for a channel that includes a nonlinear element of limited bandwidth, such as an LED. Multilevel Pulse Amplitude Modulation (PAM) was selected as the signal format, as well as a variant of the Volterra series equalizer as the compensating element. The results obtained may be used for the correction of the dynamic characteristics of LEDs applied in VLC systems. For the sake of comparison, we used Modulation Error Ratio (MER) values at the receiver output. The dynamic nonlinear behavior was modeled by a Wiener–Hammerstein device, whereas the post/pre-equalizer was based on the dynamic deviation reduction-based Volterra series. The obtained results indicate that the post- and pre-equalizer performed comparably for the linear/moderately nonlinear channels and for a high noise level. In the case of high nonlinearity and a large SNR (Signal–to–Noise Ratio) values, the post-equalizer performed somewhat better in terms of MER by a few dB at maximum.

## 1. Introduction

Visible Light Communications (VLC) have gained much popularity lately [1] and are treated as a complementary addition to 5G/6G wireless systems. In the VLC system, a white Light-Emitting Diode (LED) plays a double role as a light source and a transmitter in the downstream direction. The main problem with this solution is that the LED modulation bandwidth is rather limited, as well as its current/power characteristic exhibits dynamic nonlinearity [2]. Thus, in order to obtain sufficient throughputs, one needs to use multivalued/multilevel modulation formats, such as PAM (Pulse Amplitude Modulation), CAP (Carrierless Amplitude Phase), and DMT (Discrete MultiTone) [3], as well as compensate for the LED frequency response and its nonlinearity [4]. For the latter, various equalizers are used [4,5,6,7,8,9]. Among them, there are equalizers based on the general Hammerstein model [6], decision feedback equalizer (DFE) with static nonlinearity compensation [8], equalizers based on the Volterra and Wiener approach [4,5], devices based on neural networks [7], and others [9]. Usually, the equalizer is employed at the receiver side (post-equalizer), but there are solutions that insert the compensating device at the transmitter (pre-equalizer) [10,11]. Immediately, a question arises as to which solution (pre- or post-equalization) is better. Such a question may be readily answered only in a simple case of a linear system and zero-forcing equalizer [12]. In this case, i.e., identical powers transmitted in both cases, a linear channel, and a zero-forcing (ZF) equalizer, the value of the signal–to–noise ratio (SNR) at the receiver is identical in both cases (pre- and post-equalization) [12]. According to the so-called p-th order inverse theory [13], some types of nonlinear equalizers should work equally well in both configurations (pre and post) in the noiseless environment [13]. Unfortunately, the situation is not so clear for other equalizer types and for specific nonlinear channels in the presence of noise.

The present paper compares the pre- and post-equalizer performances in terms of the received signal quality for a channel that includes a nonlinear element of limited bandwidth, such as LED. We selected multilevel Pulse Amplitude Modulation (PAM) as the signal format due to its relative simplicity. There were many types of equalizers to choose from for our comparison, but we selected the so-called reduction-based Volterra series equalizer [13,14]. One reason for that was that it was general enough and able to compensate for dynamic nonlinearity [3]; the other reason was that the number of coefficients used (computation complexity) was much lower as compared to the “classical” Volterra series approach [14]. The nonlinear element (e.g., LED) was modeled by the Wiener–Hammerstein device [15]. The selection of such a model followed from its relative simplicity and ease in separately varying its bandwidth and nonlinearity, as well as from the fact that there is no direct relation between the coefficients of the Wiener–Hammerstein and Volterra models. The comparison of the pre- and post-equalization schemes was conducted in [16] for direct current-biased optical–orthogonal frequency division multiplexing (DCO–OFDM). However, in such a system, the pre-/post-equalization is understood simply as the channel gain division between the transmitter and receiver. Therefore, the results of [16], although interesting, cannot be directly applied here.

The rest of this work is organized as follows. The Section 2 presents the setup of the analyzed system, as well as gives some analytical background for the simulation. The Section 3 describes the obtained results, while the Section 4 discusses and tries to explain them. The Section 5 concludes the presented research.

## 2. Simulation Setup

In this section, we shall present the setup used in our simulations. A block schematic of the modeled transmission system is shown in Figure 1 for the training mode. 

A PAM signal with 2, 4, 8, and 16 levels was employed. The respective data symbols (with 2, 4, 8, and 16 levels depending on the modulation) are generated in a pseudo-random way and convolved with a root raised cosine (RRC) filter, whose impulse response is given by
(1)$$h\left(t\right)=\{\begin{array}{c} 1+\alpha \left(\frac{4}{\pi }-1\right)\;for\;\;t=0\\ \frac{\alpha }{\sqrt{2}}\left[\left(1+\frac{2}{\pi }\right)sin\frac{\pi }{4\alpha }+\left(1-\frac{2}{\pi }\right)cos\frac{\pi }{4\alpha }\right]\;\mathrm{for}\;t=\pm \frac{T}{4\alpha }\;\\ \frac{sin\left[\pi \frac{t}{T}\left(1-\alpha \right)\right]+4\alpha \frac{t}{T}cos\left[\pi \frac{t}{T}\left(1+\alpha \right)\right]}{\pi \frac{t}{T}\left[1-{\left(4\alpha \frac{t}{T}\right)}^{2}\right]}\;\;\;\;\;\;\;\;\mathrm{elsewhere} \end{array}$$
where *T* is the symbol period and *α* is the so-called roll-off factor. We used over-sampling with 10 samples per transmitted symbol. The signal after convolution was fed to a device with dynamic nonlinearity, which simulated the nonlinear element, such as an LED, in the transmission channel. A block schematic of this device is depicted in Figure 2.

It consists of two linear low pass filters of the first order, and the memoryless polynomial nonlinearity is sandwiched between these filters. The impulse responses of the linear filters, $g_{1,2}$, are given by
(2)$$g_{1,2}(t)=\left\{\begin{array}{c} \exp\left(-\frac{t}{\tau_{1,2}}\right)\mathrm{for}\;t\geq 0\\ 0\;\;\;\;\;\;\;\mathrm{elsewhere} \end{array}\right.$$
where *τ_1_, τ_2_* are the time constants of the first and second filter, respectively. For the sake of simplification, the results presented in the sequel are shown for the identical time constants of both filters, *τ_1_* = *τ_2_* = *τ*.

In order to provide the same power of the input signals to the nonlinear elements (polynomial nonlinearity here, and the equalizer), the average powers of the signals at the filter outputs are normalized to unity. Between the linear filters, there is the memoryless device in the form of the 4th-order polynomial. The output of this polynomial, *w*, is related to its input, *v*, via
(3)$$w=av+bv^{2}+cv^{3}+dv^{4}$$
where *a*, *b*, *c*, and *d* are the nonlinearity coefficients of the respective order.

It readily follows from the description and Figure 2 that the dynamic nonlinearity is modeled by a cascade of Wiener and Hammerstein elements [15]. In the training mode, when the coefficients of the equalizer are calculated, white noise is added to the normalized signal outgoing from the Wiener–Hammerstein element, as shown in Figure 1. Namely, to the subsequent samples of this signal, independent realizations of a random variable are added. This random variable has the Gaussian probability density function (pdf), zero mean, and σ^2^ variance. Changing this variance, we may control the signal–to–noise ratio (*SNR*) at the receiver input. Since the signal power is normalized to unity, we have
(4)$$SNR=-10log_{10}\sigma^{2}$$

The signals with noise are input to the nonlinear equalizer based on the so-called modified Volterra series [14], which reduces the complexity of the Volterra approach. If we denote the input signal to the equalizer by *x*(*i*) the output of the employed equalizer, *y*(*i*), is given by
(5)$$y\left(i\right)=a_{1}x\left(i\right)+a_{2}x^{2}\left(i\right)+a_{3}x^{3}\left(i\right)+\overset{30}{\underset{k=1}{\sum }}a_{k+3}\left[x\left(i-k\right)-x\left(i\right)\right]\;+x\left(i\right)\overset{30}{\underset{k=1}{\sum }}a_{k+33}\left[x\left(i-k\right)-x\left(i\right)\right]+x^{2}\left(i\right)\overset{30}{\underset{k=1}{\sum }}a_{k+63}\left[x\left(i-k\right)-x\left(i\right)\right]$$
where $a_{k}$ (*k* = 1, …, 93) are equalizer coefficients.

The coefficient [$a_{k}$] is calculated in a recurrent training procedure using an error signal *e*(*i*)
(6)e(i)=Y(i)−z(i−10)
where the error signal *e*(*i*) is the difference between the signal *Y*(*i*), which comes from the reverse RRC filter fed from the equalizer output, *y*(*i*), and the output of the inverse RRC filter, *z*(*i-10*), working in an ideal case (no noise and no nonlinear distortion). See Figure 1 for the configuration. The subtraction in the argument of z is caused by the processing delay (in this case, ten samples) in the equalizer. Let us denote by *X*(*i*) the signal vector, defined as
(7)$$X\left(i\right)=[x\left(i\right),\;x^{2}\left(i\right),x^{3}\left(i\right),\left\{\;x\left(i-1\right)-x\left(i\right)\right\},\left\{x\left(i-2\right)-x\left(i\right)\right\},\ldots \{x\left(i-30\right)\;\begin{array}{c} \;-x\left(i\right)\},x\left(i\right)\left\{x\left(i-1\right)-x\left(i\right)\right\},x\left(i\right)\{x\left(i-2\right)\\ \;-x\left(i\right)\},\ldots x\left(i\right)\left\{x\left(i-30\right)-x\left(i\right)\right\},x^{2}\left(i\right)\{x\left(i-1\right)\\ \;-x\left(i\right)\},\;x^{2}\left(i\right)\left\{x\left(i-2\right)-x\left(i\right)\right\},\ldots x^{2}\left(i\right)\left\{x\left(i-30\right)-x\left(i\right)\right\}] \end{array}$$

Then, the equalizer coefficients [$a_{k}$] are modified in each step according to a reccurential formula
(8)$$\left[a_{k}\left(i\right)\right]=\left[a_{k}\left(i-1\right)\right]-0.0001\times e\left(i\right)\times \frac{X\left(i\right)}{\left|X\left(i\right)\right|},\;k=1,\;\ldots ,\;93$$
where |*X*(*i*)| is the Euclidean norm of *X*(*i*).

At the simulation’s next stage, the equalizer with the coefficients calculated as described above is used to boost the signal quality either at the receiver (as a post-equalizer), see Figure 3, or at the transmitter (as a pre-equalizer), see Figure 4. In either case, the data set used to assess the performance of the pre/post-equalizer is completely separate from the training data set. In order to quantitively assess the performance of both equalizers, we used the modulation error ratio (*MER*) parameter slightly tailored to the current situation. It is defined here as
(9)$$MER=10log_{10}\frac{\sum_{k=1}^{N}{\left(z_{k}^{\prime }\right)}^{2}}{\sum_{k=1}^{N}{\left(Y_{k}^{\prime }-z_{k}^{\prime }\right)}^{2}}\;\;\;\left[dB\right]$$
where *Y’_k_* is the signal value at the RRC filter output at the moment of sampling (after pre- or post-correction was applied), whereas *z’_k_* is the respective value of the ideal signal (no distortions and no noise). Taking into account that, due to normalization, the mean value of the ideal signal is unity, we finally have
(10)$$MER=-10\;log_{10}\left[\frac{1}{N}\overset{N}{\underset{k=1}{\sum }}{\left(Y_{k}^{\prime }-z_{k}^{\prime }\right)}^{2}\right]$$

## 3. Results

The schematic of the system simulating the PAM signal transmission, which was presented in the previous section, was used for the comparison of MER values obtained when the equalizer was used either at the transmitter or at the receiver. The equalizer coefficients were calculated separately in each case, i.e., were suited to given values of the PAM constellation, type of nonlinearity, noise level, roll-off factor, and filter time constant. In any case, the data/noise sequences used for training and measurements were ~100 k symbols long. Separate data were always used for training and measurement. Simulations were conducted for three nonlinearity variants (see Equation (3)) beginning with the linear system.

[a,b,c,d] = [1, 0, 0, 0], through moderate nonlinearity [a,b,c,d] = [1, −0.1, 0.05, 0.02] to strong nonlinearity [a,b,c,d] = [1, 0.3, −0.15, −0.06]. The time constant in the Wiener–Hammerstein element was also changed, as well as the roll-off factor, α. Exemplary eye patterns obtained for the moderate nonlinearity, roll-off factor α = 0.3, τ = 0.3 T (T is the symbol time duration), and SNR = 20 dB are depicted in Figure 5, Figure 6, Figure 7 and Figure 8 for all PAM modulations applied. One can readily observe that, in the ideal case (lack of noise and nonlinearity), the separation of transmitted symbols is possibly independent of the number of modulation levels. However, if the distortions are added, the correct reception is possible only for PAM-2, unless the equalizer is applied. The application of a pre/post-equalizer considerably improved the eye patterns for two, four, and eight modulation levels, which is particularly visible for PAM-4. However, for PAM-16, even the post/pre-equalizer does not give an open eye pattern. For the used parameters set, it appears that the post-equalizer performs slightly better than the pre-equalizer.

Figure 9, Figure 10 and Figure 11 present the values of MER obtained for linear, moderately nonlinear, and highly nonlinear elements, respectively. They are shown for all modulations used (PAM-2, …, PAM-16), roll-off factors α = 0.1 or 1, time constants in the Wiener–Hammerstein element τ = 0.3T or T, and receiver signal–to–noise ratios SNR = 0 dB, 10 dB, or 20 dB. The MER values for pre- and post-equalization are depicted and also the MER values when no equalization is applied for the sake of comparison.

Let us notice at the beginning that the MER values for PAM-4, PAM-8, and PAM-16 were very close to one another. This did not fully apply to PAM-2, whose MER values were slightly different in some cases. The obtained results indicate that, for any type of linear/nonlinear channel, the improvement of MER obtained by the equalizer depended primarily on the receiver SNR; the best results (the greatest MER improvement) were observed for the greatest SNR value at the receiver (20 dB) and the worse results (the least MER improvement) were obtained for the smallest SNR (0 dB). In the linear case, the MER improvement was around 15, …, 20 dB for SNR = 20 dB, and dropped to 2, …, 7 dB for SNR = 0 dB. For moderate nonlinearity, the respective improvements were 12, …, 17 dB (SNR = 20 dB), and 2, …, 7 dB (SNR = 0 dB). At last, for strong nonlinearity, we had 5, …, 15 dB (SNR = 20 dB), and 2, …, 5 dB (SNR = 0 dB). The increase in the noise level (SNR decrease) caused MER reduction for any type of equalization. In the linear case, the reduction was 10, …, 15 dB when the SNR decreased from 20 dB to 0 dB. For the moderate nonlinearity, the corresponding reduction was around 6, …, 13 dB, and, for the strong nonlinearity, the maximum MER reduction was around 11 dB. The change in the time constant in the Wiener–Hammerstein filter from 0.3 T to T caused the respective reduction of the channel bandwidth, and, as the consequence, the MER decreased, no matter if any equalization was used. This reduction varied between 3 dB and 9 dB, and was the greatest for the maximum SNR at the receiver. The nonlinearity diminished the receiver SNR’s impact on MER. In turn, the changes in the roll-off factor, α, from 0.1 to 1 only slightly affected the obtained results. In the linear case, such a change in α caused the MER to reduce by 2 dB at maximum. Similar values were obtained for moderate nonlinearity, whereas, for the strong nonlinearity, the results were similar for both α = 0.1 and α = 1, and they were even better (larger MER) for the latter in some cases (PAM-2). The comparison of the system performance for pre- and post-equalization is the most interesting. Generally, one may state that both equalization methods gave comparable results in terms of MER. The pre-equalization performed better for the linear case and a small SNR (maximum improvement of 2 dB compared to post-equalization), whereas the post-equalization was better for a large SNR and high nonlinearity (maximum improvement of 8 dB compared to pre-equalization). However, apart from the case of strong nonlinearity and SNR = 20 dB, the pre- and post-equalization performed similarly within 2 dB of MER.

## 4. Discussion

It is necessary to stress that the above comparison between pre- and post-equalization was conducted for a particular case of a nonlinear device in the form of the Wiener–Hammerstein element consisting of the memoryless fourth-order polynomial sandwiched between two low-pass filters of the first order. Moreover, the equalizer under consideration had reduced complexity and was not equivalent to the general Volterra filter. Bearing these in mind, we cannot state that the obtained results are general. However, they may indicate how pre- and post-equalizers perform for specific nonlinear channels with restricted bandwidths. We start our discussion with the most obvious results. As is well known, the MER value depends on two factors: noise and inter-symbol interference (ISI). The latter is caused by the signal transmission through a cascade of filters (RRC filter, Wiener–Hammerstein element, equalizer, and reverse RRC filter) that is both nonlinear and has bandwidth restrictions. It is obvious that the noise increase (e.g., by 10 dB) caused an MER reduction, but it was respectively lower (in this case less than 10 dB) due to the ISI source of error, which does not change with the SNR. In the same way, the reduction of the channel bandwidth via the increase in the time constant in the Wiener–Hammerstein device, as well as the boost of nonlinearity itself, increased the ISI, which led to the MER reduction. Therefore, it is visible in the figures that the MER value for τ = T is worse than for τ = 0.3T. Also, when nonlinearity is increased and other parameters are kept constant, it results in MER reduction. It is obvious that the equalizer of any type cannot fully compensate for the distortions, although it improves the MER value. Generally speaking, bandwidth reduction and nonlinearity are competing effects that cause ISI. Thus, the change in the Wiener–Hammerstein filters’ time constant (change in device bandwidth) affected the MER less for the high nonlinearity than for the linear case. In a similar way, the change in nonlinearity affected the MER more when the Wiener–Hammerstein element bandwidth was greater (smaller τ) than when the bandwidth was reduced (greater τ), as the nonlinearity was the dominant effect in the former case. The same reasons explain the fact that the higher the SNR value, the more visible the influence of the remaining parameters (τ, nonlinearity, etc.) on MER. On the other hand, the roll-off factor, α, did not much affect MER. This may be readily explained by remembering that a change in α does not alter the (noise) channel bandwidth and did not alter the principle that the signal passing by a matched pair of RRC filters does not have ISI at the moments of sampling.

The obtained MER values were similar for pre-and post-equalization and the linear/moderately nonlinear channel and/or a high noise level (low SNR). In this case, the MER value is mostly determined by the small-signal (linear) bandwidth restrictions, and then (linear channel and ZF equalization) [12] the pre- and post-equalization are equivalent. When the nonlinear distortions are dominant (high nonlinearity, large SNR), slightly better results are obtained for post-equalization. At a first glance, it appears that it contradicts the result presented in [13] stating that the order of connecting the dynamic nonlinear element and its equalizer in a cascade does not affect the outcome (equalization quality). However, the above holds only for the so-called p-th order Volterra representation, whereas, in this work, we do not use the general Volterra series, but its simplification (dynamic deviation reduction-based Volterra series). Besides, the work [13] dealt with noiseless signals, and, here, the noise is a substantial factor to be considered.

The results here are also somewhat different from those obtained in [16], where pre-equalization performed somewhat better than post-equalization. However, in [16], another modulation (DCO–OFDM) was used, the channel model was different, and pre-/post-equalization meant simply the channel gain division between the transmitter and receiver. Thus, direct comparison is not possible, but it only stresses the necessity of further research. It should include an actual dynamic model of nonlinear elements (e.g., LED), as well as involve other modulation schemes, such as discrete multitone (DMT), and a carrier-less amplitude phase (CAP). One should consider also the extension of the equalizer model.

## 5. Conclusions

This work was devoted to the comparison of the performance of pre- and post-equalization in a noisy system with dynamic nonlinearity with memory and limited bandpass. The results obtained may be used for the correction of the dynamic characteristics of LEDs applied in VLC systems. For the sake of comparison, we used MER values at the receiver output. The dynamic nonlinear behavior was modeled by a Wiener–Hammerstein device, whereas the post/pre-equalizer was based on the dynamic deviation reduction-based Volterra series. The obtained results indicate that the post- and pre-equalizer performed comparably for the linear/moderately nonlinear channel and for a high noise level. In the case of high nonlinearity and large SNR values, the post-equalizer performed somewhat better in terms of MER by a few dB at maximum.

Another topic that was not treated here is the dependence of the equalization schemes on the interference from other VLC connections. The issue of light interference in VLC is a complex one, and the reader is referred to [17] for a review. If there are many such interfering connections, we may treat them as independent random processes and their sum has an approximately Gaussian probability density function. In this case, the interference simply reduces the SNR at the receiver. On the other hand, in the presence of one or few predominant interfering sources, this is no longer true, and further research is needed.

## Figures and Tables

**Figure 1 sensors-22-01782-f001:**
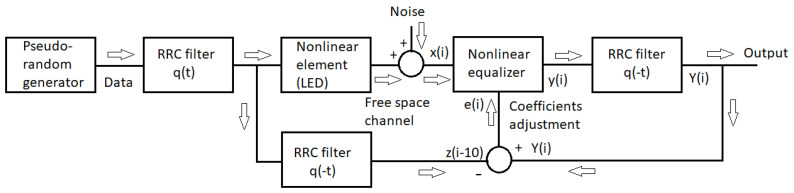
Block schematic of the transmission system in the training mode.

**Figure 2 sensors-22-01782-f002:**
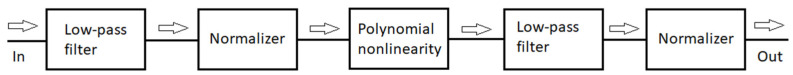
Block schematic of the Wiener–Hammerstein filter modeling the dynamic nonlinearity.

**Figure 3 sensors-22-01782-f003:**
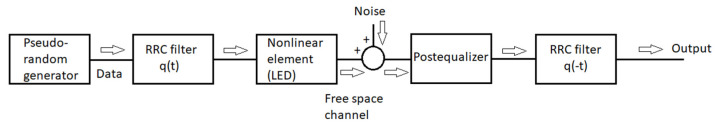
Post-equalizer configuration.

**Figure 4 sensors-22-01782-f004:**
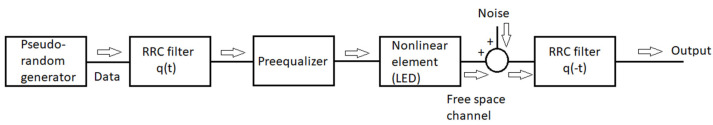
Pre-equalizer configuration.

**Figure 5 sensors-22-01782-f005:**
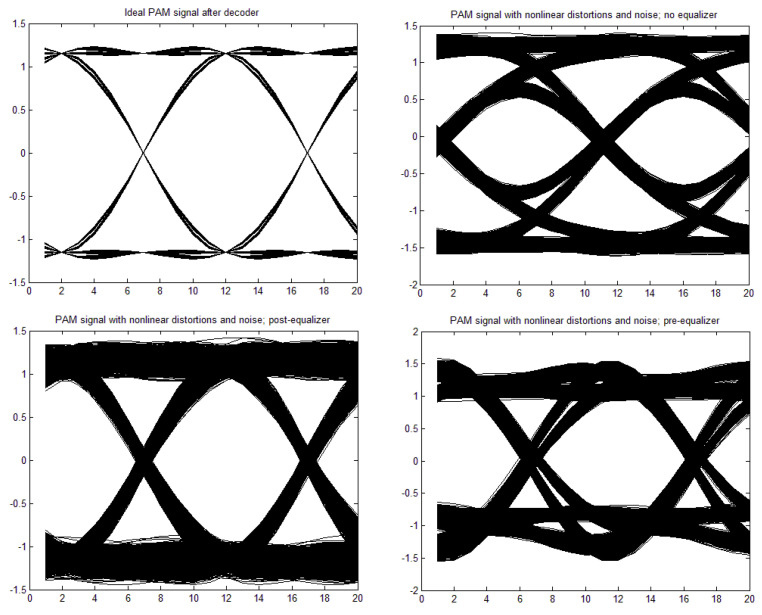
Eye patterns for PAM-2. Moderate nonlinearity: [1, −0.1, 0.05, 0.02], and α = 1, SNR = 20 dB, and τ = 0.3T.

**Figure 6 sensors-22-01782-f006:**
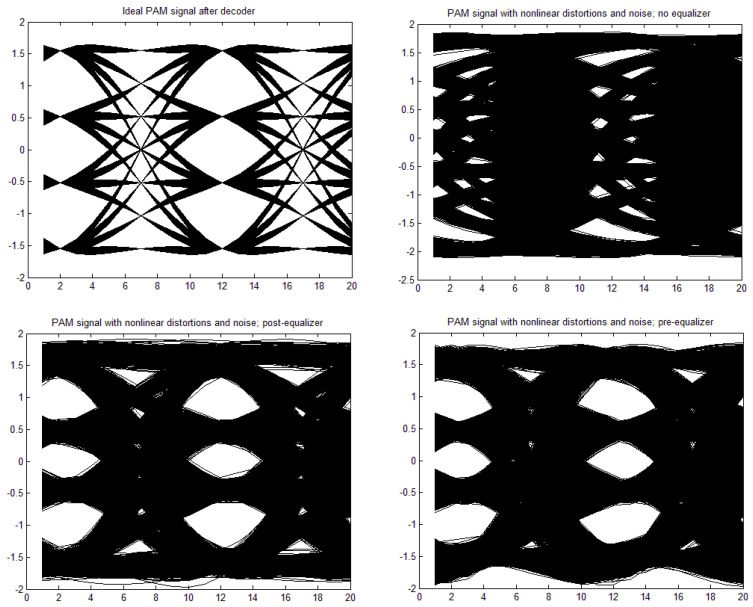
Eye patterns for PAM-4. Moderate nonlinearity: [1, −0.1, 0.05, 0.02], and α = 1, SNR = 20 dB, and τ = 0.3T.

**Figure 7 sensors-22-01782-f007:**
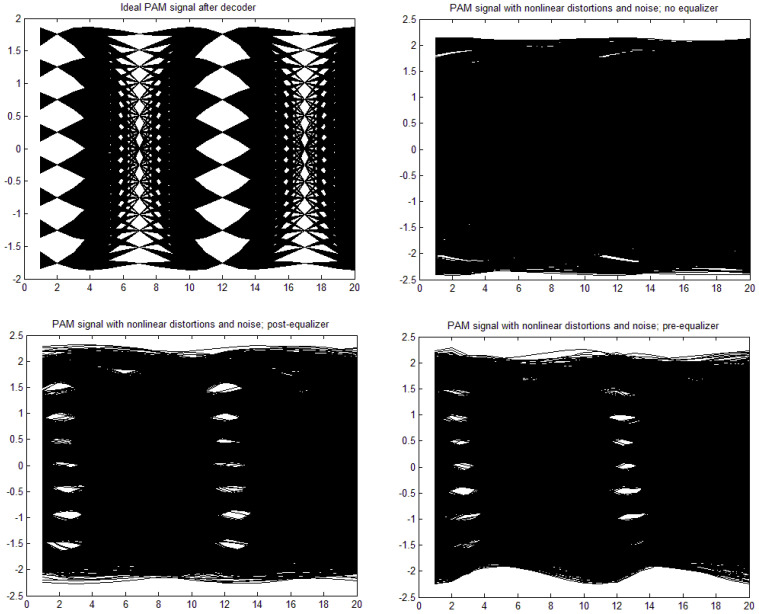
Eye patterns for PAM-8. Moderate nonlinearity: [1, −0.1, 0.05, 0.02], and α = 1, SNR = 20 dB, and τ = 0.3T.

**Figure 8 sensors-22-01782-f008:**
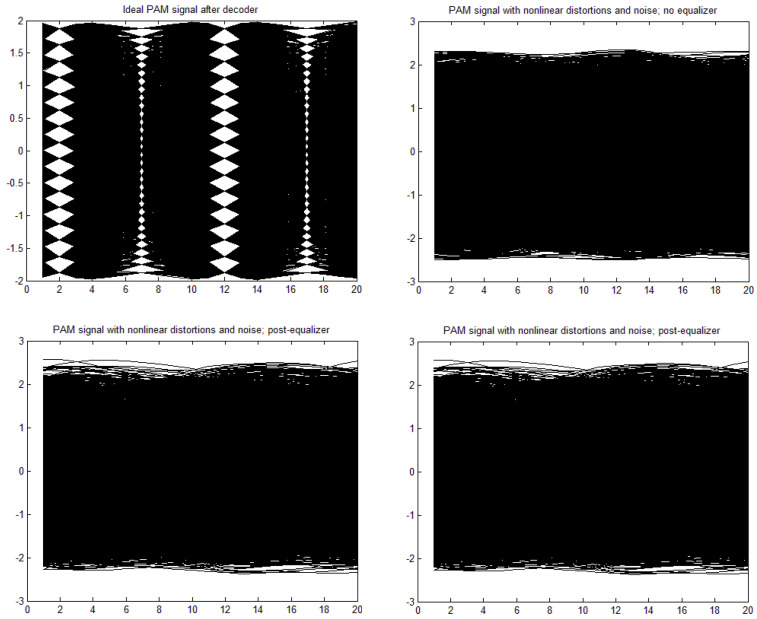
Eye patterns for PAM-16. Moderate nonlinearity: [1, −0.1, 0.05, 0.02], and α = 1, SNR = 20 dB, and τ = 0.3T.

**Figure 9 sensors-22-01782-f009:**
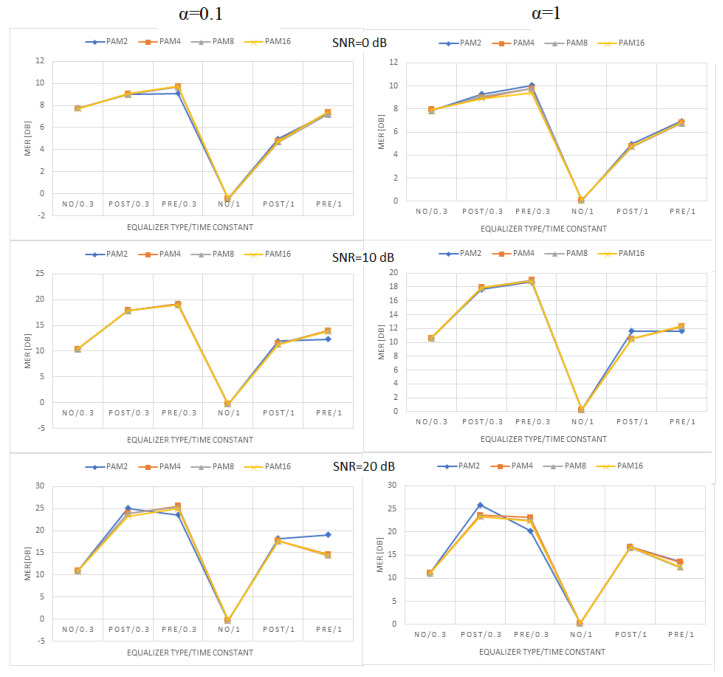
MER values obtained in the linear case ([1, 0, 0, 0]).

**Figure 10 sensors-22-01782-f010:**
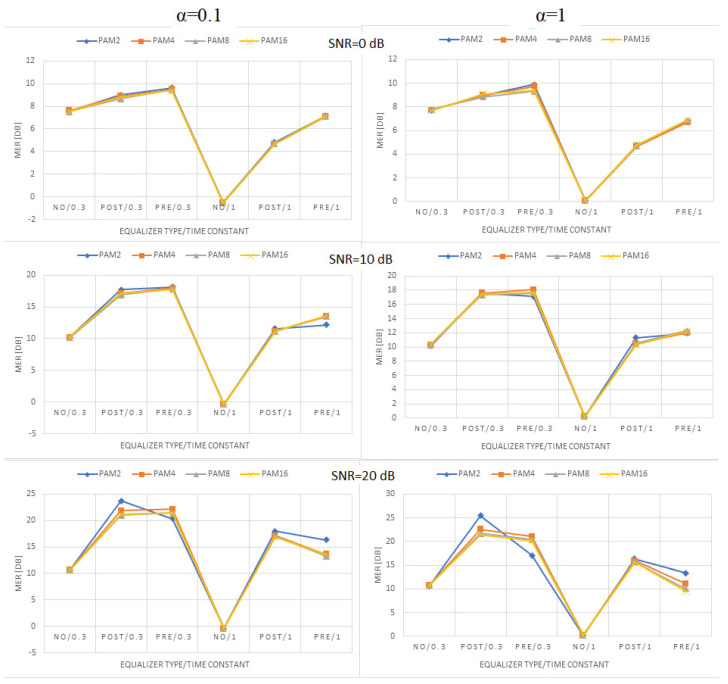
MER values obtained for the moderate nonlinearity ([1, −0.1, 0.05, 0.02]).

**Figure 11 sensors-22-01782-f011:**
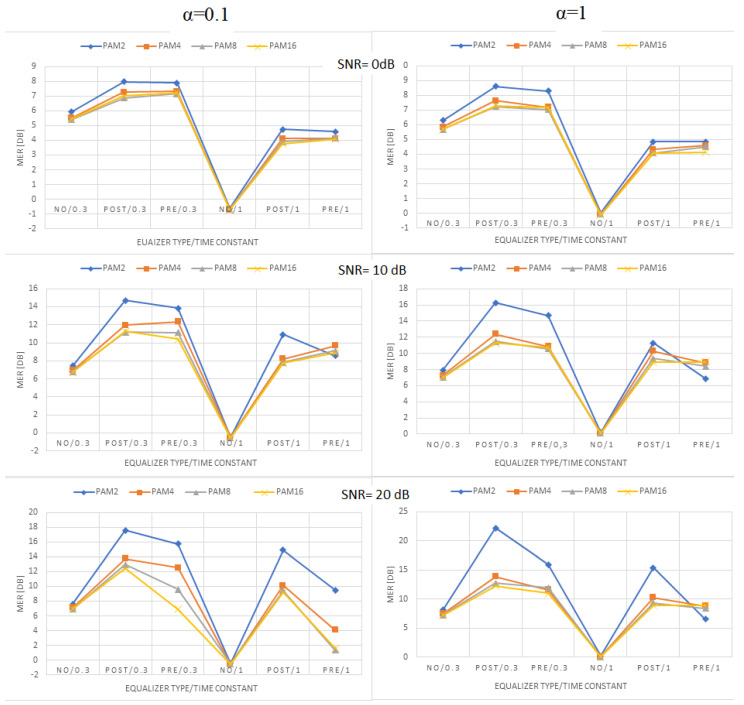
MER values obtained for high nonlinearity ([1, 0.3, −0.15, −0.06]).

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
