# Peer review of "Comparison of the Nonlinear Dynamic Pre- and Post-LED Equalization"

_sensors, 2022, doi:10.3390/s22051782_

Round 1

Reviewer 1 Report

Considering that visible light communication is attracting great interest in the field of next generation cellular networks, the presented work on LED nonlinear dynamic equalization in the form of Wiener-Hammerstein element and Volterra series based on dynamic deviation reduction may be of importance in this field. I recommend publication of this paper if some clarifications and improvements are made:
1) All acronyms (standard and defined by the author) should be defined at the first mention. For example, the acronym LED (Light-Emitting Diode) is never explained in this manuscript. I suggest that the explanation of LED should also be given in the abstract and introduction. The acronym MER (modulation error ratio) should also be defined in the abstract. 
2) On page 2, line 73, use the spaces between the numbers. 
3) I suggest that Figure 1 be made clearer and relate to the text. For example, the text mentions the pseudo-random generator, but there is no such device in the figure. Another point that confuses the reader is that in the text it is mentioned that LED is represented by a device with dynamic nonlinearity, but in the figure there is the element "Nonlinearity with memory". Also, it would be nice if the authors indicated the free space channel in Figure 1. The same improvements are needed in Figures 3 and 4. 
4) On page 4, line 169, the space between "100" and "k" is missing.
5) On page 8, line 216, the space at 20dB is missing.
6) On page 8, line 221, the "dB" after "SNR=0" is missing. 
7) From the results of the article, it appears that the post and pre equalizers perform comparably for linear/moderately nonlinear channels and at high noise levels, but it would be nice if the authors could also comment on the dependence on interference from other VLC connections. Researching interference can be very complex, but perhaps the authors could elaborate a bit or even mention it as further research at the end of the article.

Author Response

Dear Sirs,

At the beginning let me thank the reviewers for their comments which vastly improve the paper quality. Below please find the comments with my amendments marked in bold.

Reviewer 1

“Considering that visible light communication is attracting great interest in the field of next generation cellular networks, the presented work on LED nonlinear dynamic equalization in the form of Wiener-Hammerstein element and Volterra series based on dynamic deviation reduction may be of importance in this field. I recommend publication of this paper if some clarifications and improvements are made:

1) All acronyms (standard and defined by the author) should be defined at the first mention. For example, the acronym LED (Light-Emitting Diode) is never explained in this manuscript. I suggest that the explanation of LED should also be given in the abstract and introduction. The acronym MER (modulation error ratio) should also be defined in the abstract. 

All acronyms are now defined when used for the first time both in the main text and in the abstract.

2) On page 2, line 73, use the spaces between the numbers. 

Done.

3) I suggest that Figure 1 be made clearer and relate to the text. For example, the text mentions the pseudo-random generator, but there is no such device in the figure. Another point that confuses the reader is that in the text it is mentioned that LED is represented by a device with dynamic nonlinearity, but in the figure there is the element "Nonlinearity with memory". Also, it would be nice if the authors indicated the free space channel in Figure 1. The same improvements are needed in Figures 3 and 4. 

Corrected as requested in all three figures: The block description “Data” was replaced with “Pseudo-random generator”. The block description “Nonlinearity with memory” was replaced with “Nonlinear element (LED)”. The free space channel was indicated in all three figures.

4) On page 4, line 169, the space between "100" and "k" is missing.

The space was inserted.

5) On page 8, line 216, the space at 20dB is missing.

The space was inserted.

6) On page 8, line 221, the "dB" after "SNR=0" is missing. 

dB was added.

7) From the results of the article, it appears that the post and pre equalizers perform comparably for linear/moderately nonlinear channels and at high noise levels, but it would be nice if the authors could also comment on the dependence on interference from other VLC connections. Researching interference can be very complex, but perhaps the authors could elaborate a bit or even mention it as further research at the end of the article.”

A suitable passage was added at the end of the conclusions section as well as a reference [17] that reviews the topic.

I have to stress that the paper formatting has changed due to the editing mode applied.

Kind regards,

J. Siuzdak

Reviewer 2 Report

The paper compares the pre- and post-equalizer performance regarding LEDs used in Visible Light Communication (VCL) links, given that such LEDs exhibit low bandwidth and non-linear behavior.

The paper presents its case in an adequate and scientifically sound manner. There seem to be differences with results presented in other works but the author gives explanation for that.

There is some overlapping between sections 4 and 5 (e.g. 244-245 with 289-290 and 276-277 with 295-296).

I would suggest comments regarding ref. [16] to move to section 4 (Discussion) and session 5 (Conclusions) only contained conclusions regarding the present work.

The text needs moderate editing regarding the use of English.

I consider the paper publishable subject to comments above.

Author Response

Dear Sirs,

At the beginning let me thank the reviewers for their comments which vastly improve the paper quality. Below please find the comments with my amendments marked in bold.

Reviewer 2

The paper compares the pre- and post-equalizer performance regarding LEDs used in Visible Light Communication (VCL) links, given that such LEDs exhibit low bandwidth and non-linear behavior.

The paper presents its case in an adequate and scientifically sound manner. There seem to be differences with results presented in other works but the author gives explanation for that.

There is some overlapping between sections 4 and 5 (e.g. 244-245 with 289-290 and 276-277 with 295-296).

In order to reduce this ovelapping the first sentence in section 4 was removed (original 244-245) and the following one modified as well as the sentence (original 276-277) was modified.

I would suggest comments regarding ref. [16] to move to section 4 (Discussion) and session 5 (Conclusions) only contained conclusions regarding the present work.

The comments regarding ref. [16] were moved from section 5 to section 4 as requested.

The text needs moderate editing regarding the use of English.

I am unable to perform the language editing on such a short notice (5 days) but if necessary it can be done after the paper acceptance.

I consider the paper publishable subject to comments above.”

I have to stress that the paper formatting has changed due to the editing mode applied.

Kind regards,

J. Siuzdak